# OpenReview forum: "Input Compensation for Pruned Models"
_ICLR.cc/2025/Conference — Submitted to ICLR 2025_

### Official Review · Reviewer_dTty · 2024-10-28

**Soundness:** 2
**Presentation:** 2
**Contribution:** 1
**Rating:** 5
**Confidence:** 3

**Summary:**

[Update]

Thanks for the authors' feedback. My questions have been answered in the authors' feedback. I can see some value of this work, based on its improvements over baseline methods on the reported benchmark datasets. However, the proposed method can be framed as a special LoRA method, to some extent. Although LoRA is applied to model weight, for a deep network the weight compensation in the current layer is actually the "Input Compensation" to the next layer. So it is hard to claim a strong novelty for this work.

Based on the above consideration, I would like to keep my initial vote.

----- Original Review -----

In their research, the authors introduced a new approach to enhance the precision of pruned large language models. This innovative technique involves compensating the input with a learned $\Delta_x$ value, which is calculated using a learnable attention-like linear layer. The method necessitates fine-tuning on the target dataset. To establish the effectiveness of their proposed approach, the authors conducted experiments on various popular vision and language assessment datasets.

**Strengths:**

* The authors presented comprehensive numerical evaluations on popular benchmarks and compared it with multiple prominent baseline methods, including magnitude pruning techniques, Wanda, and Sparse GPT.

* The proposed technique can be utilized in both vision and language models, as it is agnostic to the specific model architecture. This allows for versatility in its application across various domains of artificial intelligence and machine learning.

**Weaknesses:**

* The proposed method for Input Compensation (IC) has some limitations in terms of novelty. The primary idea is similar to weight compensation techniques, except it diverges from such approaches by focusing on input space. It is noteworthy that an IC method can be transformed into a weight compensation method, as the two strategies are "dual" to one another.

* While the reported improvements seem to be promising, it is essential to consider their fairness in comparison to the baseline methods. The proposed IC method necessitates re-training and fine-tuning on the target dataset, which may introduce bias in the evaluation results if the baseline methods were not similarly fine-tuned. In such cases, it would be reasonable to conclude that the proposed technique's performance could potentially be attributed to the difference in fine-tuning processes rather than solely due to its inherent merits.


* It is important to note that the proposed IC method does introduce additional learnable parameters, which might not be considered when comparing the models’ sparsity levels. This could lead to an inaccurate estimation of the actual sparsity levels of both the original model and the proposed technique. Therefore, it is crucial to consider these extra parameters when evaluating the sparsity levels and overall performance of the models in question.

**Questions:**

* If the baseline models were not fine-tuned in the comparison, please report the fine-tuned version of these models / methods.
* When computing the sparsity level, is the additional learnable parameters counted in the IC-related numbers?

---

> ### Author Response · Authors · 2024-11-24
> **Reply to Reviewer dTty (1/n)**
>
> We sincerely thank the reviewer for the valuable comments on our work. We address the raised concerns as follows. If there are any remaining questions, please let us know and we are happy to address them.
>
> ---
>
> > **Q1.**
> The proposed method for Input Compensation (IC) has some limitations in terms of novelty. The primary idea is similar to weight compensation techniques, except it diverges from such approaches by focusing on input space. It is noteworthy that an IC method can be transformed into a weight compensation method, as the two strategies are "dual" to one another.
>
> **A1.**
> Thanks for your insightful comments.
> Input compensation and weight compensation are different as compared below:
>
> (i) Note that duality between input and weight space only holds for the **linear** layer, i.e.,
> for an input compensation $\Delta X$, we can design a weight compensation $\Delta W\equiv X^{-1}\Delta X W$ (assume $X$ is invertible) such that:
> $$(X+\Delta X)W= XW + \Delta X W=XW+XX^{-1}\Delta X W=X(W+X^{-1}\Delta X W)=X(W+\Delta W).$$
> However, for **nonlinear** models $\mathcal{F}(X;W)$,
> the duality does not always hold, i.e., for an input compensation $\Delta X$, we cannot find $\Delta W$ such that
> $$\mathcal{F}(X+\Delta X; W) = \mathcal{F}(X; W+\Delta W).$$
>
> (ii) Weight compensation is **input-independent**; however,
> input compensation is input-dependent.
> Input-dependent compensation is more flexible.
>
> (iii) For input compensation, we edit the input without modifying the model; thus, we can serve **one shared model for all tasks**.
> For weight compensation, however, we need to serve a **task-specific model** for each task.
> Hence, for multiple tasks, weight compensation needs much more serving resources than input compensation.
>
> (iv) Input compensation is **complementary** to weight compensation. Thus, **input compensation can be integrated into weight compensation**, as shown by the successful applications of Magnitude + IC, Wanda + IC, and SparseGPT + IC.
>
>
> ---
>
> **Q2.**
> > While the reported improvements seem to be promising, it is essential to consider their fairness in comparison to the baseline methods. The proposed IC method necessitates re-training and fine-tuning on the target dataset, which may introduce bias in the evaluation results if the baseline methods were not similarly fine-tuned. In such cases, it would be reasonable to conclude that the proposed technique's performance could potentially be attributed to the difference in fine-tuning processes rather than solely due to its inherent merits.
> >
> > If the baseline models were not fine-tuned in the comparison, please report the fine-tuned version of these models / methods.
>
> **A2.**
> In the paper (Lines 458-466),
> an experiment has been conducted to analyze whether input compensation is beneficial to the sparse models finetuned on the target dataset.
> As can be seen from Figure 4 of the paper,
> **IC consistently brings large improvements to existing pruning methods when sparse retraining is applied.**

---

> ### Author Response · Authors · 2024-11-24
> **Reply to Reviewer dTty (2/n)**
>
> >**Q3.**
> It is important to note that the proposed IC method does introduce additional learnable parameters, which might not be considered when comparing the models’ sparsity levels. This could lead to an inaccurate estimation of the actual sparsity levels of both the original model and the proposed technique. Therefore, it is crucial to consider these extra parameters when evaluating the sparsity levels and overall performance of the models in question.
> >
> >When computing the sparsity level, is the additional learnable parameters counted in the IC-related numbers?
>
> **A3.**
> The compensation pool contains **few** parameters, only 2.3M.
> To address this concern, we re-calculate the number of non-zero parameters to include the pool.
> We also test baseline methods with a lower sparsity to evaluate whether our IC performs better than baseline methods when using the same number of non-zero parameters.
> The tables below show the testing accuracy of image classification when using ViT-B/32 and ViT-B/16.
> As can be seen,
> **when using the same number of non-zero parameters, combining IC with existing pruning methods is better.**
>
>
> `Testing accuracy on image classification using CLIP ViT-B/32.`
>
> | Method | Sparsity | #Non-zero Parameters    | CIFAR100 | Flowers | Food | EuroSAT | SUN | UCF | SVHN | Pets | DTD | RESISC | Avg |
> | :----------------| :----------------: | :------: | :----: | :----: |:----: |:----: |:----: |:----: |:----: |:----: |:----: |:----: |:----: |
> | Magnitude | 50% | 110M | 33.9 | 26.1 | 34.2 | 45.6 | 30.8 | 35.4 | 45.3 | 38.7 | 27.9 | 55.4 | 37.3 |
> | Magnitude | 48% | 112M | 43.4 | 30.9 | 44.5 | 61.3 | 36.3 | 43.9 | 56.7 | 48.9 | 31.4 | 64.8 | 46.2 |
> | Magnitude + IC| 50% | 112M | 73.0 | 62.9 | 72.4 | 96.5 | 48.9 | 63.1 | 94.4 | 69.2 | 44.1 | 87.1 | **71.2** |
> ||
> | Wanda | 50% | 110M | 75.0 | 56.4 | 74.1 | 95.2 | 50.8 | 59.7 | 91.8 | 57.6 | 43.4 | 84.4 | 68.9 |
> | Wanda | 48% | 112M | 77.6 | 63.9 | 77.5 | 95.9 | 55.2 | 65.0 | 89.3 | 65.1 | 45.7 | 87.9 | 72.3 |
> | Wanda + IC| 50% | 112M | 80.1 | 76.4 | 80.4 | 97.9 | 54.7 | 69.1 | 96.1 | 77.5 | 49.8 | 91.6 | **77.4** |
> ||
> | SparseGPT | 50% | 110M | 83.3 | 69.1 | 81.6 | 97.9 | 58.0 | 68.5 | 93.7 | 59.4 | 48.2 | 89.8 | 74.9 |
> | SparseGPT | 48% | 112M | 84.5 | 72.4 | 82.7 | 97.9 | 60.4 | 71.7 | 91.2 | 66.3 | 50.2 | 91.3 | 76.9 |
> | SparseGPT + IC| 50% | 112M | 82.9 | 76.2 | 83.1 | 98.2 | 57.2 | 71.0 | 96.7 | 79.7 | 53.8 | 92.9 | **79.2** |
>
> `Testing accuracy on image classification using CLIP ViT-B/16.`
>
> | Method | Sparsity | #Non-zero Parameters    | CIFAR100 | Flowers | Food | EuroSAT | SUN | UCF | SVHN | Pets | DTD | RESISC | Avg |
> | :----------------| :----------------: | :------: | :----: | :----: |:----: |:----: |:----: |:----: |:----: |:----: |:----: |:----: |:----: |
> | Magnitude | 50% | 109M | 76.9 | 56.5 | 78.3 | 90.7 | 51.2 | 65.6 | 95.3 | 62.9 | 42.8 | 82.1 | 70.2 |
> | Magnitude | 47% | 111M | 80.9 | 64.6 | 82.1 | 94.4 | 56.5 | 70.1 | 95.9 | 68.5 | 47.3 | 86.9 | 74.7 |
> | Magnitude + IC| 50% | 111M | 82.9 | 86.7 | 84.7 | 97.6 | 60.1 | 75.1 | 97.1 | 82.5 | 61.1 | 92.8 | **82.1**|
> ||
> | Wanda | 50% | 109M | 84.1 | 78.1 | 85.5 | 97.6 | 59.5 | 68.9 | 96.9 | 72.7 | 51.8 | 91.2 | 78.6  |
> | Wanda | 47% | 111M |  85.6 | 81.7 | 86.8 | 98.0 | 62.8 | 71.7 | 96.8 | 76.8 | 55.3 | 92.9 | 80.8 |
> | Wanda + IC| 50% | 111M | 84.6 | 84.5 | 86.5 | 98.4 | 61.2 | 73.2 | 97.5 | 81.5 | 60.9 | 94.1 | **82.3**  |
> ||
> | SparseGPT | 50% | 109M | 87.2 | 80.2 | 88.1 | 98.0 | 63.8 | 73.8 | 97.0 | 75.6 | 56.4 | 93.7 | 81.4
>  |
> | SparseGPT | 47% | 111M | 87.7 | 82.3 | 88.8 | 98.3 | 63.4 | 76.2 | 97.0 | 80.4 | 59.8 | 94.6 | 82.9  |
> | SparseGPT + IC| 50% | 111M | 86.2 | 82.8 | 87.8 | 98.4 | 63.8 | 75.5 | 97.6 | 83.6 | 63.5 | 94.7 | **83.4** |

---

### Official Review · Reviewer_YvaV · 2024-10-29

**Soundness:** 2
**Presentation:** 3
**Contribution:** 2
**Rating:** 5
**Confidence:** 5

**Summary:**

This paper delves into pruning LLMs by focusing on input compensation rather than model parameter adjustment. The authors introduce a method that learns a compensation pool, consisting of multiple candidate compensations, using an attention mechanism on calibration data. This method has been tested on both classification and language tasks, demonstrating its ability to enhance the performance of pruning methods like Wanda and SparseGPT to a certain extent.

**Strengths:**

1. **Innovative Approach:** The article presents a fresh perspective on reducing reconstruction error in sparse LLMs by adjusting inputs instead of traditional parameter optimization. This approach potentially opens new avenues for LLM pruning.

2. **Theoretical Justification:** The discussion in section 4.1 provides a solid theoretical basis. By decomposing model parameters into a sparse matrix, \(S\), and a low-rank matrix, \(AB\), the authors argue convincingly that removing the \(AB\) portion and learning it back can recover performance effectively.

**Weaknesses:**

1. **Similarity to Prompt Tuning:** The method bears a resemblance to prompt tuning, with the primary difference being the integration manner (addition versus concatenation). A direct comparison between the proposed method and prompt tuning could clarify their distinct advantages and limitations.
2. **Experimental Scope:** The experiments primarily compare the proposed method with post-training sparsity methods that do not involve backpropagation. This seems limiting, as the method aligns more closely with sparse LLM fine-tuning strategies. A comparison with similar methods that also leverage fine-tuning, such as DsNot[1], would provide a more comprehensive evaluation of its effectiveness.
3. **Unclear foundation:** Although it is interesting to model parameters as a low-rank matrix plus a sparse matrix, how exactly is the sparsity of \( S \) defined? Does it mean that all levels of sparsity (50%, 60%, 70%) are compatible with such a formulation?  Furthermore, there are many possibilities for deciding sparse matrices. Can all types of sparse matrices be adequately represented by adding a low-rank matrix (as the proposed IC method)?

[1] Dynamic Sparse No Training: Training-Free Fine-tuning for Sparse LLMs. In ICLR, 2024.

**Questions:**

Please see the weakness part.

---

> ### Author Response · Authors · 2024-11-24
> **Reply to Reviewer YvaV (1/n)**
>
> We sincerely thank the reviewer for the valuable comments on our work. We address the raised concerns as follows. If there are any remaining questions, please let us know and we are happy to address them.
>
> ---
>
> > **Q1.**
> Similarity to Prompt Tuning: The method bears a resemblance to prompt tuning, with the primary difference being the integration manner (addition versus concatenation). A direct comparison between the proposed method and prompt tuning could clarify their distinct advantages and limitations.
>
> **A1.**
> As discussed in Lines 127-129 of the paper, input compensation is **different** from prompting:
> - Prompting inserts extra tokens into the input, but input compensation edits the input directly. Hence, **no extra tokens** are added. Note that adding tokens increases the computational cost.
> - Prompts are input-independent; however, our designed input compensation is input-dependent. Note that **input-dependent compensation is more flexible than input-independent prompting**.
>
> ---
>
> > **Q2.**
> Experimental Scope: The experiments primarily compare the proposed method with post-training sparsity methods that do not involve backpropagation. This seems limiting, as the method aligns more closely with sparse LLM fine-tuning strategies. A comparison with similar methods that also leverage fine-tuning, such as DsNot[1], would provide a more comprehensive evaluation of its effectiveness.
>
> **A2.**
> Thanks for your suggestions. We conducted an additional comparison between DSnoT [1] and DSnoT + IC.
> The table below shows the WikiText perplexity for the LLaMA model family with 50% sparsity.
> As can be seen, Wanda + DSnoT + IC consistently outperforms Wanda + DSnoT, validating that **IC improves the performance of finetuned sparse LLMs**.
>
>
> |   Method                   | Sparsity | LLaMA-1 (7B) | LLaMA-2 (7B) | LLaMA-3.1 (8B) |
> |----------------------|----------|--------------|--------------|----------------|
> | Wanda + DSnoT        | 50%      | 7.14         | 6.40         | 9.01           |
> | Wanda + DSnoT + IC   | 50%      | **7.05**     | **6.24**     | **8.69**       |
>
> ---
>
> > **Q3.**
> Unclear foundation: Although it is interesting to model parameters as a low-rank matrix plus a sparse matrix, how exactly is the sparsity of ( S ) defined? Does it mean that all levels of sparsity (50%, 60%, 70%) are compatible with such a formulation? Furthermore, there are many possibilities for deciding sparse matrices. Can all types of sparse matrices be adequately represented by adding a low-rank matrix (as the proposed IC method)?
>
> **A3.**
> We believe the reviewer comments on Lines 209-211, which says the weight matrix can be approximately decomposed into a sparse matrix $\mathbf{S}$ and a low-rank matrix.
> Which type of sparse matrix can be used to perform this decomposition is not the focus of this paper.
> As studied in existing works (Yu et al., 2017; Li et al., 2023b; Ding et al., 2023), a very high sparsity (e.g., 70%) leads to weak performance.
>
> Note that this decomposition is only used to motivate our attention-based compensation mechanism in the linear layer and is **NOT an assumption** when applying input compensation to pruned models.

---

### Official Review · Reviewer_Pwcr · 2024-11-01

**Soundness:** 3
**Presentation:** 2
**Contribution:** 2
**Rating:** 5
**Confidence:** 4

**Summary:**

This paper presents an approach to improve pruned model performance through input compensation, offering a new perspective by optimizing the input space instead of the parameter space. Experiments and analysis on several baseline methods and benchmarks validate the effectiveness of the proposed input compensation approach.

**Strengths:**

- The paper is easy to follow, and the input compensation (IC) method is clearly stated.
- The method is evaluated across several tasks including image classification, language modeling, and image generation.
- The performance gain of the proposed IC over the CLIP ViT-B model on several baseline methods like SparseGPT and Wanda is notable.

**Weaknesses:**

- While the main goal of network pruning is to reduce computational costs, the introduction of an encoder and attention-based compensation mechanism appears to add more computational overhead during inference. It would be better for the authors to clearly quantify the additional computation and memory costs introduced by the compensation mechanism.
- The optimization process for the compensation pool needs a more detailed explanation, as the approaches described in Sections 5.1 and 5.2 seem quite different.
- The evaluation lacks testing on the widely adopted structured 2:4 sparsity setting, which is an important baseline for practical applications.
- The sparsity calculation needs clarification, especially considering the additional parameters introduced by the compensation pool and the encoder.
- It would be better to list the detailed environment for better reproducibility including hardware and software.

**Questions:**

- It is stated in Sec 5.1 of the paper that the compensation pool is shared across all tasks. How does the compensation vary across different input types and model architectures?
- In Table 4, only the result under 70% sparsity setting is reported. How's the performance of applying input compensation under the 50% sparsity setting? It would be better to directly compare with the result reported in the original Wanda paper for fair comparison.

---

> ### Author Response · Authors · 2024-11-24
> **Reply to Reviewer Pwcr (1/n)**
>
> We sincerely thank the reviewer for the valuable comments on our work. We address the raised concerns as follows. If there are any remaining questions, please let us know and we are happy to address them.
>
> ---
>
> > **Q1.**
> While the main goal of network pruning is to reduce computational costs, the introduction of an encoder and attention-based compensation mechanism appears to add more computational overhead during inference. It would be better for the authors to clearly quantify the additional computation and memory costs introduced by the compensation mechanism.
>
> **A1.**
> Thanks for the comments.
> We agree that running the encoder of IC introduces extra computation costs.
> However, the cost is **low** when the encoder is small.
> As summarized in the table below (image classification experiment with CLIP ViT-B/32), compared with Magnitude, the computation cost of Magnitude + IC increases by **only 1%** (from 305G to 309G FLOPs) and inference speed drops by **only 20%** (from 665 to 535 images/second).
> Note that the **testing accuracy increases largely** (as shown in Table 1 of the paper), verifying that this additional computation cost is **worthwhile**.
>
> | Method | Sparsity    | Computation Cost (FLOPs) | Inference Speed (images/second) | memory (G) |
> | :----------------| :----------------: | :------: | :----: | :----: |
> | Magnitude | 50% | 305 G  | 665 | 3.7 |
> | Magnitude + IC| 50% | 309 G | 535 | 3.7 |
>
> ---
>
>
> > **Q2.**
> The optimization process for the compensation pool needs a more detailed explanation, as the approaches described in Sections 5.1 and 5.2 seem quite different.
>
>
> **A2.** To address this concern, we clarify the optimization details below.
>
> - For image classification experiments (Section 5.1), $\mathbf{v}_i$'s are learnable pixels on all sides (with the padding size 30). The compensation pool $\mathbf{K}$ and $\mathbf{V}$ are randomly initialized by standard normal distribution. We train the pool for 30 epochs using the SGD optimizer with a learning rate of 40 and momentum of 0.9, while the mini-batch size is 128. We use the supervised loss Eq.(5) and reuse the pruned image encoder as the encoder of IC.
>
> - For NLP experiments (Section 5.2), $\mathbf{K}$ and $\mathbf{V}$ are learnable embeddings, which are randomly initialized by a normal distribution with zero mean and standard deviation 0.01. We train the pool for 20 epochs using the AdamW optimizer with a learning rate of 0.001 and a linear warmup schedule. The mini-batch size is 2. We use the reconstruction loss Eq.(7) and reuse the LLM's input embedding layer as the encoder of IC.
>
> ---
>
> > **Q3.**
> The evaluation lacks testing on the widely adopted structured 2:4 sparsity setting, which is an important baseline for practical applications.
>
> **A3.** Thanks for your suggestions.
> We conducted additional image classification experiments using CLIP ViT-B/32 and ViT-B/16 to evaluate whether IC is useful for structured 2:4 sparsity.
> The tables below show the testing accuracy.
> We can see that IC consistently brings large improvements to existing pruning methods, demonstrating that **IC is effective when using structured 2:4 sparsity**.
>
> `Testing accuracy on image classification tasks using CLIP ViT-B/32.`
> | Method | Sparsity    | CIFAR100 | Flowers | Food | EuroSAT | SUN | UCF | SVHN | Pets | DTD | RESISC | Avg |
> | :----------------| :----------------: | :------: | :----: | :----: |:----: |:----: |:----: |:----: |:----: |:----: |:----: |:----: |
> | Magnitude | 2:4 |  30.7 | 11.2 | 19.8 | 42.7 | 19.4 | 23.3 | 27.5 | 25.8 | 15.6 | 35.3 | 25.1 |
> | Magnitude + IC| 2:4 | 78.6 | 66.5 | 77.9 | 97.4 | 53.9 | 67.5 | 96.1 | 70.9 | 44.3 | 90.2 | **74.3** |
> ||
> | Wanda | 2:4 | 39.6 | 14.5 | 35.1 | 46.1 | 21.7 | 25.5 | 42.4 | 25.2 | 20.9 | 41.4 | 31.2 |
> | Wanda + IC| 2:4 | 80.6 | 74.5 | 80.7 | 97.7 | 54.9 | 68.1 | 96.5 | 74.1 | 51.4 | 91.6 | **77.0** |
> ||
> | SparseGPT | 2:4 | 75.5 | 40.2 | 73.0 | 94.3 | 44.5 | 52.6 | 61.3 | 45.7 | 33.5 | 81.6 | 60.2 |
> | SparseGPT + IC| 2:4 | 82.2 | 79.4 | 82.6 | 98.4 | 57.7 | 69.9 | 96.8 | 76.4 | 54.3 | 92.5 | **79.0** |
>
> `Testing accuracy on image classification tasks using CLIP ViT-B/16.`
> | Method | Sparsity    | CIFAR100 | Flowers | Food | EuroSAT | SUN | UCF | SVHN | Pets | DTD | RESISC | Avg |
> | :----------------| :----------------: | :------: | :----: | :----: |:----: |:----: |:----: |:----: |:----: |:----: |:----: |:----: |
> | Magnitude | 2:4 |  68.3 | 39.9 | 64.5 | 78.6 | 39.4 | 53.2 | 95.3 | 56.9 | 32.9 | 67.6 | 59.7 |
> | Magnitude + IC| 2:4 | 83.9 | 83.8 | 85.9 | 97.9 | 61.4 | 75.7 | 97.4 | 84.0 | 57.3 | 93.3 | **82.1** |
> ||
> | Wanda | 2:4 | 71.5 | 46.4 | 71.0 | 89.7 | 40.6 | 50.4 | 96.2 | 61.4 | 33.5 | 75.2 | 63.6 |
> | Wanda + IC| 2:4 | 84.9 | 87.2 | 86.9 | 98.2 | 62.7 | 75.4 | 97.4 | 84.3 | 59.2 | 94.1 | **83.0** |
> ||
> | SparseGPT | 2:4 | 82.8 | 66.2 | 82.8 | 94.4 | 53.6 | 63.0 | 97.2 | 74.2 | 43.1 | 90.2 | 74.7 |
> | SparseGPT + IC | 2:4 | 85.6 | 89.6 | 87.7 | 98.4 | 63.9 | 76.9 | 97.6 | 84.9 | 61.2 | 94.9 | **84.1** |

---

> ### Author Response · Authors · 2024-11-24
> **Reply to Reviewer Pwcr (2/n)**
>
> > **Q4.**
> The sparsity calculation needs clarification, especially considering the additional parameters introduced by the compensation pool and the encoder.
>
> **A4.**
> Thanks for the valuable comments.
> Our proposed method reuses the submodule of the pruned model as the encoder and only requires a **very small** compensation pool as the additional parameters.
> For example, in image classification, IC reuses the pruned image encoder as the encoder and the compensation pool contains only 2.3M parameters.
> In natural language processing, IC reuses the input embedding layer of LLM as the encoder and designs a tiny compensation pool with 2.6M parameters.
>
> To account for these additional pool parameters, we recalculate the number of non-zero parameters in IC.
> For a fair comparison, we also test the baseline methods using the same number of non-zero parameters.
> The tables below show the testing accuracy on image classification tasks using CLIP ViT-B/32 and ViT-B/16. As can be seen, **IC outperforms all the baseline methods using the same number of non-zero parameters**, validating the usefulness of our method again.
>
> `Testing accuracy on image classification using CLIP ViT-B/32.`
>
> | Method | Sparsity | #Non-zero Parameters    | CIFAR100 | Flowers | Food | EuroSAT | SUN | UCF | SVHN | Pets | DTD | RESISC | Avg |
> | :----------------| :----------------: | :------: | :----: | :----: |:----: |:----: |:----: |:----: |:----: |:----: |:----: |:----: |:----: |
> | Magnitude | 50% | 110M | 33.9 | 26.1 | 34.2 | 45.6 | 30.8 | 35.4 | 45.3 | 38.7 | 27.9 | 55.4 | 37.3 |
> | Magnitude | 48% | 112M | 43.4 | 30.9 | 44.5 | 61.3 | 36.3 | 43.9 | 56.7 | 48.9 | 31.4 | 64.8 | 46.2 |
> | Magnitude + IC| 50% | 112M | 73.0 | 62.9 | 72.4 | 96.5 | 48.9 | 63.1 | 94.4 | 69.2 | 44.1 | 87.1 | **71.2** |
> ||
> | Wanda | 50% | 110M | 75.0 | 56.4 | 74.1 | 95.2 | 50.8 | 59.7 | 91.8 | 57.6 | 43.4 | 84.4 | 68.9 |
> | Wanda | 48% | 112M | 77.6 | 63.9 | 77.5 | 95.9 | 55.2 | 65.0 | 89.3 | 65.1 | 45.7 | 87.9 | 72.3 |
> | Wanda + IC| 50% | 112M | 80.1 | 76.4 | 80.4 | 97.9 | 54.7 | 69.1 | 96.1 | 77.5 | 49.8 | 91.6 | **77.4** |
> ||
> | SparseGPT | 50% | 110M | 83.3 | 69.1 | 81.6 | 97.9 | 58.0 | 68.5 | 93.7 | 59.4 | 48.2 | 89.8 | 74.9 |
> | SparseGPT | 48% | 112M | 84.5 | 72.4 | 82.7 | 97.9 | 60.4 | 71.7 | 91.2 | 66.3 | 50.2 | 91.3 | 76.9 |
> | SparseGPT + IC| 50% | 112M | 82.9 | 76.2 | 83.1 | 98.2 | 57.2 | 71.0 | 96.7 | 79.7 | 53.8 | 92.9 | **79.2** |
>
> `Testing accuracy on image classification using CLIP ViT-B/16.`
>
> | Method | Sparsity | #Non-zero Parameters    | CIFAR100 | Flowers | Food | EuroSAT | SUN | UCF | SVHN | Pets | DTD | RESISC | Avg |
> | :----------------| :----------------: | :------: | :----: | :----: |:----: |:----: |:----: |:----: |:----: |:----: |:----: |:----: |:----: |
> | Magnitude | 50% | 109M | 76.9 | 56.5 | 78.3 | 90.7 | 51.2 | 65.6 | 95.3 | 62.9 | 42.8 | 82.1 | 70.2 |
> | Magnitude | 47% | 111M | 80.9 | 64.6 | 82.1 | 94.4 | 56.5 | 70.1 | 95.9 | 68.5 | 47.3 | 86.9 | 74.7 |
> | Magnitude + IC| 50% | 111M | 82.9 | 86.7 | 84.7 | 97.6 | 60.1 | 75.1 | 97.1 | 82.5 | 61.1 | 92.8 | **82.1**|
> ||
> | Wanda | 50% | 109M | 84.1 | 78.1 | 85.5 | 97.6 | 59.5 | 68.9 | 96.9 | 72.7 | 51.8 | 91.2 | 78.6  |
> | Wanda | 47% | 111M |  85.6 | 81.7 | 86.8 | 98.0 | 62.8 | 71.7 | 96.8 | 76.8 | 55.3 | 92.9 | 80.8 |
> | Wanda + IC| 50% | 111M | 84.6 | 84.5 | 86.5 | 98.4 | 61.2 | 73.2 | 97.5 | 81.5 | 60.9 | 94.1 | **82.3**  |
> ||
> | SparseGPT | 50% | 109M | 87.2 | 80.2 | 88.1 | 98.0 | 63.8 | 73.8 | 97.0 | 75.6 | 56.4 | 93.7 | 81.4
>  |
> | SparseGPT | 47% | 111M | 87.7 | 82.3 | 88.8 | 98.3 | 63.4 | 76.2 | 97.0 | 80.4 | 59.8 | 94.6 | 82.9  |
> | SparseGPT + IC| 50% | 111M | 86.2 | 82.8 | 87.8 | 98.4 | 63.8 | 75.5 | 97.6 | 83.6 | 63.5 | 94.7 | **83.4** |
>
> ---
>
> > **Q5.**
> It would be better to list the detailed environment for better reproducibility including hardware and software.
>
> **A5.**
> Our experiments are conducted on A100 (80G) with CUDA version 12.6.
> We list the detailed software environment (package==version) as follows.
> ```
> accelerate==0.27.2
> apex==0.9.10dev
> bitsandbytes==0.41.1
> debug_utils==1.0
> evaluate==0.4.3
> fairscale==0.4.13
> huggingface_hub==0.23.4
> intel_extension_for_pytorch==2.5.0
> lm_eval==0.4.4
> numpy==1.24.4
> optuna==4.1.0
> packaging==24.2
> peft==0.11.1
> pytorch_utils==0.5.5
> ray==2.39.0
> safetensors==0.4.5
> timm==0.9.8
> torch==2.1.2+cu118
> torch_xla==2.5.1
> torchvision==0.16.2+cu118
> tqdm==4.66.1
> transformers==4.46.2
> ```

---

> ### Author Response · Authors · 2024-11-24
> **Reply to Reviewer Pwcr (3/n)**
>
> > **Q6.**
> It is stated in Sec 5.1 of the paper that the compensation pool is shared across all tasks. How does the compensation vary across different input types and model architectures?
>
> **A6.**
> For different input types and model architectures,
> the compensations are always learnable $\mathbf{K}$ and $\mathbf{V}$.
> Compensation is **model-agnostic** and only depends on the input types:
> For an input with **continuous** values (e.g., images in Section 5.1), compensation is added to the input directly;
> For an input with **discrete** values (e.g., sentences in Section 5.2), compensation is added to its input embeddings.
>
>
> ---
>
> > **Q7.**
> In Table 4, only the result under 70% sparsity setting is reported. How's the performance of applying input compensation under the 50% sparsity setting? It would be better to directly compare with the result reported in the original Wanda paper for fair comparison.
>
> **A7.**
> Thanks for your insightful suggestion.
> We conducted additional experiments to evaluate IC under the 50% sparsity accordingly.
> The table below shows the WikiText validation perplexity of the pruned LLaMA family of models (results with $^\dagger$ are reproduced by us, while results with $^\star$ are reported in the original publication).
> As can be seen, IC consistently brings a significant improvement to existing pruning methods, verifying **the effectiveness of compensating inputs for pruned LLMs again**.
>
>
> |                  | Sparsity | LLaMA-1 (7B) | LLaMA-2 (7B) | LLaMA-3.1 (8B) |
> |------------------|----------|--------------|--------------|----------------|
> | Magnitude$^\star$       | 50%      | 17.29        | 14.89        | -         |
> | Magnitude$^\dagger$        | 50%      | 17.29        | 14.90        | 134.26         |
> | Magnitude + IC   | 50%      | **14.60**    | **12.44**    | **80.37**      |
> ||
> | Wanda$^\star$       | 50%      | 7.26         | 6.42         | -           |
> | Wanda$^\dagger$            | 50%      | 7.26         | 6.42         | 9.88           |
> | Wanda + IC       | 50%      | **7.17**     | **6.33**     | **8.80**       |
> ||
> | SparseGPT$^\star$        | 50%      | 7.22         | 6.51         | -           |
> | SparseGPT$^\dagger$        | 50%      | 7.22         | 6.52         | 9.46           |
> | SparseGPT + IC   | 50%      | **7.07**     | **6.38**     | **8.51**       |

---

### Official Review · Reviewer_6Ljo · 2024-11-01

**Soundness:** 2
**Presentation:** 3
**Contribution:** 2
**Rating:** 5
**Confidence:** 4

**Summary:**

This paper proposes an input compensation method for pruned models. which builds a compensation pool and learns the pool by minimizing the reconstruction loss. The authors empirically demonstrate the effectiveness of IC across extensive NLP and CV tasks.

**Strengths:**

The manuscript proposes an interesting idea that it uses input compensation to minimize the accuracy loss caused by weight pruning. The proposed IC shows good accuracy improvement on extensive experiments. In-depth analysis is conducted to investigate key components of IC.

**Weaknesses:**

1. The major concern is the proposed method may introduce a significant amount of additional computation cost, resulting longer inference latency. For CLIP models, while the authors claim that the pruned image encoder is used as the encoder of IC, as illustrated in Fig.1, the proposed method has to run through an additional Encoder(ViT), which is time-consuming. The authors should add the comparisons of computation cost and inference speed.
2. For structured sparsity experiments, following [1], using 2:4 pattern instead of 4:8 pattern may be better, as advanced GPUs support 2:4 pruning acceleration.
3. The proposed method is not clearly explained. More implementation details should be added.

**Questions:**

1. For CLIP experiments, the authors mentioned 'The compensation pool is shared across all ten tasks'. I suppose the pool is learned on one big dataset?
2. It would be better to provide the training time.
3. For NLP experiments, the authors claimed that they follow OWL[2] and focused on the case of 70\% unstructured sparsity. While the base pruning methods may fail at 70\% sparsity, it will be more fair to compare the case like 'wanda+owl' vs 'wanda+owl+IC'. Could IC be well combined with OWL?

[1] Mishra, Asit, et al. "Accelerating sparse deep neural networks." *arXiv preprint arXiv:2104.08378* (2021).
[2] Yin, Lu, et al. "Outlier weighed layerwise sparsity (owl): A missing secret sauce for pruning llms to high sparsity." *arXiv preprint arXiv:2310.05175* (2023).

---

> ### Author Response · Authors · 2024-11-24
> **Reply to Reviewer 6Ljo (1/n)**
>
> We sincerely thank the reviewer for the valuable comments on our work. We address the raised concerns as follows. If there are any remaining questions, please let us know and we are happy to address them.
>
> ---
>
> > **Q1.**
> The major concern is the proposed method may introduce a significant amount of additional computation cost, resulting longer inference latency. For CLIP models, while the authors claim that the pruned image encoder is used as the encoder of IC, as illustrated in Fig.1, the proposed method has to run through an additional Encoder(ViT), which is time-consuming. The authors should add the comparisons of computation cost and inference speed.
>
> **A1.**
> Thanks for the comments.
> We agree that running the encoder of IC introduces extra computation costs.
> However, the cost is **low** when the encoder is small.
> As summarized in the table below (image classification experiment with CLIP ViT-B/32), compared with Magnitude, the computation cost of Magnitude + IC increases by **only 1%** (from 305G to 309G FLOPs), and the inference speed drops by **only 20%** (from 665 to 535 images/second).
> Note that the **testing accuracy increases largely** (as shown in Table 1 of the paper), verifying that this additional computation cost is **worthwhile**.
>
> | Method | Sparsity    | Computation Cost (FLOPs) | Inference Speed (images/second) | memory (G) |
> | :----------------| :----------------: | :------: | :----: | :----: |
> | Magnitude | 50% | 305 G  | 665 | 3.7 |
> | Magnitude + IC| 50% | 309 G | 535 | 3.7 |
>
> ---
>
> > **Q2.**
> For structured sparsity experiments, following [1], using 2:4 pattern instead of 4:8 pattern may be better, as advanced GPUs support 2:4 pruning acceleration.
>
> **A2.**
> Thanks for your suggestions.
> We conducted additional image classification experiments using CLIP ViT-B/32 and ViT-B/16 to evaluate whether IC is useful for structured 2:4 sparsity.
> The tables below show the testing accuracy.
> We can see that IC consistently brings large improvements to existing pruning methods, demonstrating that **IC is effective when using structured 2:4 sparsity**.
>
> `Testing accuracy on image classification tasks using CLIP ViT-B/32.`
> | Method | Sparsity    | CIFAR100 | Flowers | Food | EuroSAT | SUN | UCF | SVHN | Pets | DTD | RESISC | Avg |
> | :----------------| :----------------: | :------: | :----: | :----: |:----: |:----: |:----: |:----: |:----: |:----: |:----: |:----: |
> | Magnitude | 2:4 |  30.7 | 11.2 | 19.8 | 42.7 | 19.4 | 23.3 | 27.5 | 25.8 | 15.6 | 35.3 | 25.1 |
> | Magnitude + IC| 2:4 | 78.6 | 66.5 | 77.9 | 97.4 | 53.9 | 67.5 | 96.1 | 70.9 | 44.3 | 90.2 | **74.3** |
> ||
> | Wanda | 2:4 | 39.6 | 14.5 | 35.1 | 46.1 | 21.7 | 25.5 | 42.4 | 25.2 | 20.9 | 41.4 | 31.2 |
> | Wanda + IC| 2:4 | 80.6 | 74.5 | 80.7 | 97.7 | 54.9 | 68.1 | 96.5 | 74.1 | 51.4 | 91.6 | **77.0** |
> ||
> | SparseGPT | 2:4 | 75.5 | 40.2 | 73.0 | 94.3 | 44.5 | 52.6 | 61.3 | 45.7 | 33.5 | 81.6 | 60.2 |
> | SparseGPT + IC| 2:4 | 82.2 | 79.4 | 82.6 | 98.4 | 57.7 | 69.9 | 96.8 | 76.4 | 54.3 | 92.5 | **79.0** |
>
> `Testing accuracy on image classification tasks using CLIP ViT-B/16.`
> | Method | Sparsity    | CIFAR100 | Flowers | Food | EuroSAT | SUN | UCF | SVHN | Pets | DTD | RESISC | Avg |
> | :----------------| :----------------: | :------: | :----: | :----: |:----: |:----: |:----: |:----: |:----: |:----: |:----: |:----: |
> | Magnitude | 2:4 |  68.3 | 39.9 | 64.5 | 78.6 | 39.4 | 53.2 | 95.3 | 56.9 | 32.9 | 67.6 | 59.7 |
> | Magnitude + IC| 2:4 | 83.9 | 83.8 | 85.9 | 97.9 | 61.4 | 75.7 | 97.4 | 84.0 | 57.3 | 93.3 | **82.1** |
> ||
> | Wanda | 2:4 | 71.5 | 46.4 | 71.0 | 89.7 | 40.6 | 50.4 | 96.2 | 61.4 | 33.5 | 75.2 | 63.6 |
> | Wanda + IC| 2:4 | 84.9 | 87.2 | 86.9 | 98.2 | 62.7 | 75.4 | 97.4 | 84.3 | 59.2 | 94.1 | **83.0** |
> ||
> | SparseGPT | 2:4 | 82.8 | 66.2 | 82.8 | 94.4 | 53.6 | 63.0 | 97.2 | 74.2 | 43.1 | 90.2 | 74.7 |
> | SparseGPT + IC | 2:4 | 85.6 | 89.6 | 87.7 | 98.4 | 63.9 | 76.9 | 97.6 | 84.9 | 61.2 | 94.9 | **84.1** |

---

> > ### Comment · Reviewer_6Ljo · 2024-11-27
> >
> > It is not clear whether the accuracy can be guranteed when the decode is small. Moreover, it is very strange that the accuracy without IC is pretty low. I will keep my rating.

---

> ### Author Response · Authors · 2024-11-24
> **Reply to Reviewer 6Ljo (2/n)**
>
> > **Q3.**
> The proposed method is not clearly explained. More implementation details should be added.
>
> **A3.**
> To address this concern, we clarify the implementation details below.
>
> - For image classification experiments (Section 5.1), $\mathbf{v}_i$'s are learnable pixels on all sides (with the padding size 30). The compensation pool $\mathbf{K}$ and $\mathbf{V}$ are randomly initialized by standard normal distribution. We train the pool for 30 epochs using the SGD optimizer with a learning rate of 40 and momentum of 0.9, while the mini-batch size is 128. We use the supervised loss Eq.(5) and reuse the pruned image encoder as the encoder of IC.
>
> - For NLP experiments (Section 5.2), $\mathbf{K}$ and $\mathbf{V}$ are learnable embeddings, which are randomly initialized by a normal distribution with zero mean and standard deviation 0.01. We train the pool for 20 epochs using the AdamW optimizer with a learning rate of 0.001 and a linear warmup schedule. The mini-batch size is 2. We use the reconstruction loss Eq.(7) and reuse the LLM's input embedding layer as the encoder of IC.
>
> ---
>
> > **Q4.**
> For CLIP experiments, the authors mentioned 'The compensation pool is shared across all ten tasks'. I suppose the pool is learned on one big dataset?
>
> **A4.**
> Yes, we merge the training sets of all ten tasks together to learn the pool.
>
> ---
>
> > **Q5.**
> It would be better to provide the training time.
>
> **A5.** The compensation pool is trained **within hours** in all experiments.
> Specifically,
> - For image classification using ViT-B/32, it takes about **2 hours** to train the compensation pool;
> - For image classification using ViT-B/16, it takes about **7 hours** to train the pool.
> - For NLP tasks, it takes about **17, 38, and 40 minutes** to train the pool for LLaMA-1 (7B), LLaMA-2 (7B) and LLaMA-3 (8B), respectively.
>
> ---
>
> > **Q6.**
> For NLP experiments, the authors claimed that they follow OWL[2] and focused on the case of 70% unstructured sparsity. While the base pruning methods may fail at 70% sparsity, it will be more fair to compare the case like 'wanda+owl' vs 'wanda+owl+IC'. Could IC be well combined with OWL?
>
> **A6.**
> Thanks for your suggestions.
> To address this concern,
> we conducted an additional experiment following OWL to combine IC with Wanda+OWL （i.e., Wanda+OWL+IC).
> The table below shows the perplexity (smaller is better).
> As can be seen, IC consistently brings improvements to Wanda+OWL by a large margin of 5.0 on all three LLaMA family of models, verifying the **effectiveness of combining IC with OWL**.
>
> | Method | Sparsity    | LLaMA-1 (7B) | LLaMA-2 (7B) | LLaMA-3.1 (8B) |
> | :----------------| :----------------: | :------: | :----: | :----: |
> | Wanda + OWL  | 70\% |   24.62 | 30.89 | 70.69 |
> | Wanda + OWL + IC |  70\% | **19.08** | **24.48** | **63.08** |

---

> ### Author Response · Authors · 2024-11-27
> **Further Reply to Reviewer 6Ljo (3/n)**
>
> We sincerely thank the reviewer for further comments.
> We are glad that our rebuttal has addressed all your initial concerns.
> For the newly raised concerns, we address them as follows.
>
> ---
>
> > **Q7.** It is not clear whether the accuracy can be guranteed when the decode is small.
>
> **A7.**
> Note that our IC method does **NOT** contain a decoder.
>
> ---
>
> > **Q8.** Moreover, it is very strange that the accuracy without IC is pretty low.
>
> **A8.** This observation (i.e., using pruning algorithms alone hurts performance) is consistent with existing works, e.g.,
>
> - Tables 2, 21, and 23 of Wanda (ICLR 2024) show that Magnitude/SparseGPT/Wanda all have **much lower testing accuracy** than the dense model;
>
> - Table 1 of SparseGPT (ICML 2023) and Table 3 of Wanda show that Magnitude/SparseGPT/Wanda have **higher perplexity** than the dense model.

---

### Author Response · Authors · 2024-11-24
**General Reply**

Dear Reviewers and ACs,

We deeply thank all the reviewers and ACs for your insightful comments and suggestions. We are delighted that reviewers find that:

- Our IC method is an **interesting** idea (`Reviewer 6Ljo`), a **fresh** perspective which opens **new** avenues for LLM pruning (`Reviewer YvaV`), a **model-agnostic** technique (`Reviewer dTty`).
- Our extensive experiments show that our IC method has **good accuracy improvement** (`Reviewer 6Ljo`), **notable** performance gain on several tasks (`Reviewer Pwcr`).
- Our analysis is **in-depth** (`Reviewer 6Ljo`) and has a **solid** theoretical basis (`Reviewer YvaV`).
- Our paper is **easy to follow** (`Reviewer Pwcr`).

We have responded to all the raised questions and hope our responses can address them. `If you have any further questions or concerns, please let us know; we will be happy to address them.`

Best,

The Authors

---

### Meta-Review · Area_Chair_88fv · 2024-12-23

**Metareview:**

The authors propose input compensation for pruned models - a method in which the input data to a pruned model is modified via additive values produced by a learned module. The learnable module consists of an attention layer with a bank of learnable values (called compensation pool). The authors show that such a modification to inputs can help improve the accuracy of pruned models.

The submission received ratings of 5, 5, 5, 5.

There are a number of issues in the experimentation which stand out:
1) A large portion of the experiment section is devoted to comparing the accuracy of untrained pruned models to the same models after the trained input compensation. This immediately stands out as inherently unfair.
2) A small part of Section 6 (Sparse Retraining with IC) compares accuracies after the pruned model is trained. The input compensation pool module is trained for 30 epochs, whereas the pruned models are trained only for 3 epochs. This is again an unfair comparison, as the pruned models might not reach their true potential. There is no evidence showing that the pruned models have saturated in their accuracy.
3) In essence, the proposed method introduces an additional learnable layer with a residual connection to the input. The total number of parameters increases by the number of keys, values, and other learnable projection layers. In the submission, there is no comparison of the increase in number of parameters or FLOPs compared to the pruned model, due to the introduction of this new module.
While there is an attempt to answer this in "Reply to Reviewer Pwcr (2/n)", the accuracies are again comparing untrained pruned networks to trained input compensation, which is an unfair comparison.

Other missing details:
- Prior work on Visual Prompt Tuning [ECCV 2022] is very related to this proposed method, and is not cited.
- In an attention layer, all input tokens interact with each other, so in some sense, even a fixed prompt affects different inputs differently due to the nature of the attention operation. There is definitely a stronger mathematical relation between fixed prompts and the method proposed here, and this should be examined more thoroughly.

Based on the above, the ACs did not find sufficient cause to overturn the negative consensus of the reviewers and recommend rejection.

**Additional Comments On Reviewer Discussion:**

The rebuttal and discussion highlighted the unfair comparisons in the submission - comparing untrained pruned models to trained input compensation models. The reviewers also drew similarities between this method and other parameter-efficient finetuning methods such as LoRA.

---

### Decision · Program_Chairs · 2025-01-22

Reject